# Rank-DETR for High Quality Object Detection

**Yifan Pu**[1]* **Weicong Liang**[2♮]* **Yiduo Hao**[3♮]* **Yuhui Yuan**[4✉]
**Yukang Yang**[4] **Chao Zhang**[2] **Han Hu**[4] **Gao Huang**[1✉]
[1]Tsinghua University  [2]Peking University  [3]University of Cambridge  [4]Microsoft Research Asia
puyf23@mails.tsinghua.edu.cn, liangweicong@stu.pku.edu.cn, yh463@cam.ac.uk,
yuhui.yuan@microsoft.com, gaohuang@tsinghua.edu.cn

## Abstract

Modern detection transformers (DETRs) use a set of object queries to predict a list of bounding boxes, sort them by their classification confidence scores, and select the top-ranked predictions as the final detection results for the given input image. A highly performant object detector requires accurate ranking for the bounding box predictions. For DETR-based detectors, the top-ranked bounding boxes suffer from less accurate localization quality due to the misalignment between classification scores and localization accuracy, thus impeding the construction of high-quality detectors. In this work, we introduce a simple and highly performant DETR-based object detector by proposing a series of rank-oriented designs, combinedly called Rank-DETR. Our key contributions include: (i) a rank-oriented architecture design that can prompt positive predictions and suppress the negative ones to ensure lower false positive rates, as well as (ii) a rank-oriented loss function and matching cost design that prioritizes predictions of more accurate localization accuracy during ranking to boost the AP under high IoU thresholds. We apply our method to improve the recent SOTA methods (e.g., H-DETR and DINO-DETR) and report strong COCO object detection results when using different backbones such as ResNet-50, Swin-T, and Swin-L, demonstrating the effectiveness of our approach. Code is available at `https://github.com/LeapLabTHU/Rank-DETR`.

## 1 Introduction

The landscape of modern object detection systems has undergone significant transformation since the pioneering work DEtection TRansformer (DETR) [3]. Since DETR yielded impressive results in object detection, numerous subsequent research works such as Deformable-DETR [78], DINO [75], and H-DETR [30] have further advanced this field. Moreover, these DETR-based approaches have been successfully extended to address various core visual recognition tasks, including instance / panoptic segmentation [6, 38, 61, 69, 71, 35, 9, 8, 70], pose estimation [57, 36, 56], and multi-object tracking [5, 47, 58]. The notable progress in these areas can be credited to ongoing advancements in enhancing the DETR-based framework for improved object detection performance.

Considerable endeavors have been dedicated to advancing the performance of DETR-based methods from various perspectives. These efforts include refining transformer encoder and decoder architectures [48, 74, 78, 2, 12, 78, 11, 40, 43] as well as redesigning of query formulations [63, 44, 34, 75]. While substantial research has been dedicated to developing accurate ranking mechanisms for dense one-stage object detectors like FCOS [59] or ATSS [77], few studies have specifically investigated this aspect for modern object detectors based on DETR. However, ranking mechanisms are vital in enhancing the average precision performance, particularly under high IoU thresholds.

---

*Equal contribution. ♮ Interns at Microsoft Research Asia.  ✉ Corresponding authors.

This study's primary focus revolves around constructing a high-quality object detector using DETR that exhibits strong performance at relatively high IoU thresholds. We acknowledge the criticality of establishing an accurate ranking order for bounding box predictions in constructing these detectors. To achieve this, we introduce two rank-oriented designs that effectively leverage the benefits of precise ranking information. First, we propose a rank-adaptive classification head and a query rank layer after each Transformer decoder layer. Rank-adaptive classification head adjusts the classification scores using rank-aware learnable logit bias vectors, while the query rank layer fuses additional ranking embeddings into the object queries. Second, we propose two rank-oriented optimization techniques: a loss function modification and a matching cost design. These functions facilitate the ranking procedure of the model and prioritize more accurate bounding box predictions with higher IoU scores when compared to the ground truth. In summary, our rank-oriented designs consistently enhance object detection performance, particularly the AP scores under high IoU thresholds.

To validate the efficacy of our approach, we conducted comprehensive experiments, showcasing consistent performance improvements across recent strong DETR-based methods such as H-DETR and DINO-DETR. For example, based on H-DETR, our method demonstrates a notable increase in $AP_{75}$ of +2.1% (52.9% vs. 55.0%) and +2.7% (55.1% vs. 57.8%) when utilizing ResNet-50 and Swin-T backbones, respectively. It is worth highlighting that our approach achieves competitive performance, reaching an 50.2% AP in the $1\times$ training schedule on the COCO `val` dataset. These results serve as compelling evidence for the effectiveness and reliability of our proposed methodology.

## 2   Related Work

**DETR for Object Detection.**   Since the groundbreaking introduction of transformers in 2D object detection by the pioneering work DETR [3], numerous subsequent studies[48, 11, 7, 63, 46] have developed diverse and advanced extensions based on DETR. This is primarily due to DETR's ability to eliminate the need for hand-designed components such as non-maximum suppression (NMS). One of the first foundational developments, Deformable-DETR [78], introduced a multi-scale deformable self/cross-attention scheme, which selectively attends to a small set of key sampling points in a reference bounding box. This approach yielded improved performance compared to DETR, particularly for small objects. Furthermore, DAB-DETR [44] and DN-DETR [34] demonstrated that a novel query formulation could also enhance performance. The subsequent work, DINO-DETR [75], achieved state-of-the-art results in object detection tasks, showcasing the advantages of DETR design by addressing the inefficiency caused by the one-to-one matching scheme. In contrast to these works, our focus lies in the design of the rank-oriented mechanism for DETR. We propose rank-oriented architecture designs and rank-oriented matching cost and loss function designs to construct a highly performant DETR-based object detector with competitive $AP_{75}$ results.

**Ranking for Object Detection.**   There exists a lot of effort to study how to improve the ranking for object detection tasks. For example, IoU-Net [31] constructed an additional IoU predictor and an IoU-guided NMS scheme that considers both classification scores and localization scores during inference. Generalized focal loss [37] proposed a quality focal loss to act as a joint representation of the IoU score and classification score. VarifocalNet [76] introduced an IoU-aware classification score to achieve a more accurate ranking of candidate detection results. TOOD [10] defined a high-order combination of the classification score and the IoU score as the anchor alignment metric to encourage the object detector to focus on high-quality anchors dynamically. In addition, ranking-based loss functions [53, 4, 49, 50, 32] are designed to encourage the detector to rank the predicted bounding boxes according to their quality and penalizes incorrect rankings. The very recent con-current work Stable-DINO [45] and Align-DETR [1] also applied the idea of IoU-aware classification score to improve the loss and matching design for DINO-DETR [75]. In contrast to the aforementioned endeavors, we further introduce a query rank scheme aimed to reduce false positive rates.

**Dynamic Neural Networks.**   In contrast to static models, which have fixed computational graphs and parameters at the inference stage, dynamic neural networks [15, 65] can adapt their structures or parameters to different inputs, leading to notable advantages in terms of performance, adaptiveness [68, 13], computational efficiency [72, 73, 67, 60], and representational power [52]. Dynamic networks are typically categorized into three types: sample-wise [27, 66, 18, 14, 51], spatial-wise [62, 28, 19, 17, 16], and temporal-wise [20, 64]. In this work, we introduce a novel query-wise dynamic approach, which dynamically integrates ranking information into the object queries based on their box quality ranking, endowing object queries with better representation ability.

## 3 Approach

Above all, we revisit the overall pipeline of the modern DETR-based methods [30, 75] in Section 3.1. The detailed design of the proposed method, including the rank-oriented architecture design and optimization design, is subsequently illustrated in Section 3.2 and Section 3.3. Eventually, we discuss the connections and differences between our approach and related works in Section 3.4.

### 3.1 Preliminary

**Pipeline.** Detection Transformers (DETRs) process an input image $\mathcal{I}$ by first passing it through a backbone network and a Transformer encoder to obtain a sequence of enhanced pixel embeddings $\mathcal{X} = \{\mathbf{x}_1, \mathbf{x}_2, \cdots, \mathbf{x}_N\}$. The enhanced pixel embeddings, along with a default set of object query embeddings $\mathcal{Q}^0 = \{\mathbf{q}_1^0, \mathbf{q}_2^0, \cdots, \mathbf{q}_n^0\}$, are then fed into the Transformer decoder. After each Transformer decoder layer, task-specific prediction heads are applied to the updated object query embeddings to generate a set of classification predictions $\mathcal{P}^l = \{\mathbf{p}_1^l, \mathbf{p}_2^l, \cdots, \mathbf{p}_n^l\}$ and bounding box predictions $\mathcal{B}^l = \{\mathbf{b}_1^l, \mathbf{b}_2^l, \cdots, \mathbf{b}_n^l\}$, respectively, where $l \in \{1, 2, \cdots, L\}$ denotes the layer index of the Transformer decoder. Finally, DETR performs one-to-one bipartite matching between the predictions and the ground-truth bounding boxes and labels $\mathcal{G} = \{\mathbf{g}_1, \mathbf{g}_2, \cdots, \mathbf{g}_m\}$ by associating each ground truth with the prediction that has the minimal matching cost and applying the corresponding supervision.

**Object Query.** To update the object query $\mathcal{Q}^0$ after each Transformer decoder layer, typically, DETRs form a total of $L$ subsets, i.e., $\{\mathcal{Q}^1, \mathcal{Q}^2, \cdots, \mathcal{Q}^L\}$, for $L$ Transformer decoder layers. For both the initial object query $\mathcal{Q}^0$ and the updated ones after each layer, each $\mathcal{Q}^l$ is formed by adding two parts: content queries $\mathcal{Q}_c^l = \{\mathbf{q}_{c,1}^l, \mathbf{q}_{c,2}^l, \cdots, \mathbf{q}_{c,n}^l\}$ and position queries $\mathcal{Q}_p^l = \{\mathbf{q}_{p,1}^l, \mathbf{q}_{p,2}^l, \cdots, \mathbf{q}_{p,n}^l\}$. The content queries capture semantic category information, while the position queries encode prior positional information such as the distribution of bounding box centers and sizes.

**Ranking in DETR.** Rank-oriented design plays a crucial role in modern object detectors, particularly in achieving superior average precision (AP) scores under high Intersection over Union (IoU) thresholds. The success of state-of-the-art detectors, such as H-DETR and DINO-DETR, relies on using simple rank-oriented designs, specifically a two-stage scheme and mixed query selection. These detectors generate the initial positional query $\mathcal{Q}_p^0$ by ranking the dense coarse bounding box predictions output by the Transformer encoder feature maps and selecting the top $\sim300$ confident ones. During evaluation, they gather $n \times K$ bounding box predictions based on the object query embedding $\mathcal{Q}^L$ produced by the final Transformer decoder layer (each query within $\mathcal{Q}^L$ generates $K$ predictions associated with each category), sort them by their classification confidence scores in descending order, and only return the top $\sim100$ most confident predictions.

In this work, we focus on further extracting the benefits brought by the ranking-oriented designs and introduces a set of improved designs to push the envelope of high-quality object detection performance. The subsequent discussion provides further elaboration on these details.

### 3.2 Rank-oriented Architecture Design: ensure lower FP and FN

While the original rank-oriented design only incorporates rank information into the initial positional query $\mathcal{Q}_p^0$, we propose an enhanced approach that leverages the benefits of sorting throughout the entire Transformer decoder process. Specifically, we introduce a rank-adaptive classification head after each Transformer decoder layer, and a query rank layer before each of the last $L-1$ Transformer decoder layers. This novel design is intended to boost the detection of true positives while suppressing false positives and correcting false negatives, leading to lower false positive rates and false negative rates. Figure 1 illustrates the detailed pipeline of our rank-oriented architecture designs.

**Rank-adaptive Classification Head.** We modify the original classification head by adding a set of learnable logit bias vectors $\mathcal{S}^l = \{\mathbf{s}_1^l, \mathbf{s}_2^l, \cdots, \mathbf{s}_n^l\}$ to the classification scores $\mathcal{T}^l = \{\mathbf{t}_1^l, \mathbf{t}_2^l, \cdots, \mathbf{t}_n^l\}$ (before $\mathrm{Sigmoid}(\cdot)$ function) associated with each object query independently. The classification predictions of the $l$-th decoder layer $\mathcal{P}^l = \{\mathbf{p}_1^l, \mathbf{p}_2^l, \cdots, \mathbf{p}_n^l\}$ can be formulated as:

$$\mathbf{p}_i^l = \mathrm{Sigmoid}(\mathbf{t}_i^l + \mathbf{s}_i^l), \quad \mathbf{t}_i^l = \mathrm{MLP}_{\mathrm{cls}}(\mathbf{q}_i^l), \tag{1}$$

where $\mathcal{Q}^l = \{\mathbf{q}_1^l, \mathbf{q}_2^l, \cdots, \mathbf{q}_n^l\}$ represents the output embedding after the $l$-th Transformer decoder layer. The hidden dimensions of both $\mathbf{t}_i^l$ and $\mathbf{s}_i^l$ are the number of categories, i.e., $K$. The overall

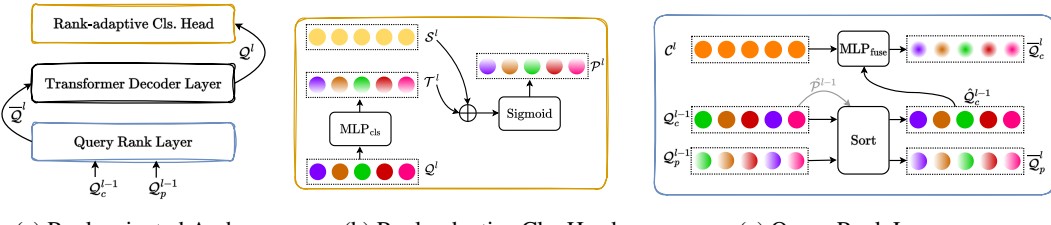

| (a) Rank-oriented Arch. | (b) Rank-adaptive Cls. Head | (c) Query Rank Layer |

Figure 1: **Illustrating the rank-oriented architecture design.** (a) The rank-oriented architecture consists of a query rank layer before each of the last $L-1$ Transformer decoder layer and a rank-adaptive classification head after each Transformer decoder layer. (b) The rank-adaptive classification head learns to adjust the classification scores accordingly. (c) The query rank layer exploits the latest ranking information to recreate the content queries and position queries used as the input to the following Transformer decoder layer.

pipeline is shown in Figure 1(b). It is noteworthy that we can directly incorporate a set of learnable embedding, denoted as $\mathcal{S}^l$, into the classification scores $\mathcal{T}^l$. This is practicable because the associated $\mathcal{Q}^l$ *has already been sorted* in the query rank layer, as explained below.

**Query Rank Layer.** We further introduce a query rank layer before each of the last $L-1$ Transformer decoder layers to regenerate the sorted positional query and content query accordingly.

First, we explain how to construct the rank-aware content query:

$$\overline{\mathcal{Q}}_c^l = \text{MLP}_{\text{fuse}}(\hat{\mathcal{Q}}_c^{l-1}\|\mathcal{C}^l), \quad \hat{\mathcal{Q}}_c^{l-1} = \text{Sort}(\mathcal{Q}_c^{l-1};\hat{\mathcal{P}}^{l-1}), \tag{2}$$

where we first sort the output of $(l-1)$-th Transformer decoder layer $\mathcal{Q}_c^{l-1}$ in descending order of $\hat{\mathcal{P}}^{l-1} = \text{MLP}_{\text{cls}}(\mathcal{Q}_c^{l-1})$. Since each element in $\hat{\mathcal{P}}^{l-1}$ is a $K$-dimensional vector, we use the maximum value over $K$ categories (classification confidence) as the ranking basis. The operation symbol $\text{Sort}(A;B)$ sorts elements within $A$ based on the decreasing order of the elements in $B$. Then, we concatenate ($\|$) the sorted object content queries $\hat{\mathcal{Q}}_c^{l-1}$ with a set of randomly initialized content query $\mathcal{C}^l$ in the feature dimension, where $l \in \{2, \cdots, L\}$. This set of content query $\mathcal{C}^l$ is optimized in an end-to-end manner. Subsequently, we fuse them back to the original dimension using a fully connected layer ($\text{MLP}_{\text{fuse}}$). In other words, for each Transformer decoder layer, we maintain a set of rank-aware static content embeddings shared across different samples. These embeddings effectively model and utilize the distribution of the most frequent semantic information [26].

Next, we present the mathematical formulations for computing the rank-aware positional query. To align the order of the positional queries with the ranked content query, we either sort or recreate the positional queries, depending on the initialization method of positional queries for different DETR-based detectors. For H-DETR, which inherits Deformable DETR and uses the same positional query for all $L$ Transformer decoder layers, we simply sort the positional query of the previous layer:

$$\overline{\mathcal{Q}}_p^l = \text{Sort}(\overline{Q}_p^{l-1};\hat{\mathcal{P}}^{l-1}), \tag{3}$$

For DINO-DETR, which generates new positional queries from the bounding box predictions in each Transformer decoder layer, we sort the bounding box predictions of each object query and recreate the positional query embedding from the sorted boxes:

$$\overline{\mathcal{Q}}_p^l = \text{PE}(\overline{\mathcal{B}}^{l-1}), \quad \overline{\mathcal{B}}^{l-1} = \text{Sort}(\mathcal{B}^{l-1};\hat{\mathcal{P}}^{l-1}), \tag{4}$$

where $\mathcal{B}^{l-1}$ and $\hat{\mathcal{P}}^{l-1}$ represent the bounding box predictions and classification predictions based on the output of $(l-1)$-th Transformer decoder layer, i.e., $\mathcal{Q}^{l-1}$. $\text{PE}(\cdot)$ contains a sine position encoding function and a small multilayer perceptron to recreate the positional query embedding $\overline{\mathcal{Q}}_p^l$. In other words, each element of $\overline{\mathcal{Q}}_p^l$ is estimated by $\overline{\mathbf{q}}_{p,i}^l = \text{PE}(\overline{\mathbf{b}}_i^{l-1})$. In Figure 1(c), we illustrate the positional query update process for H-DETR (Equation (3)) and omit that process for DINO-DETR (Equation (4)), because we primarily conducted experiments on H-DETR.

Finally, we transmit the regenerated rank-aware positional query embedding and content query embedding to the subsequent Transformer decoder layer.

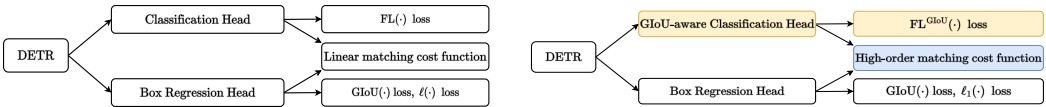

(a) The original Matching Cost and Loss          (b) Rank-oriented Matching Cost and Loss

Figure 2: **Illustrating the rank-oriented matching cost and loss design.** (a) The original DETR and its variants use a classification head and a bounding box regression head to perform predictions. The matching cost function is a linear combination of the classification scores and bounding box overlap scores. (b) The rank-oriented matching cost and loss scheme uses a GIoU-aware classification head and high-order matching cost function to prioritize predictions of more accurate localization accuracy.

**Analysis.** The key motivations behind these two rank-oriented architecture designs are adjusting the classification scores of the object queries according to their ranking order information. Within each Transformer decoder layer, we incorporate two sets of learnable representations: the logit bias vectors $\mathcal{S}^l$ and the content query vectors $\mathcal{C}^l$. By leveraging these two rank-aware representations, we have empirically demonstrated the capability of our approach to effectively address false positives ($\text{oLRP}_{\text{FP}}$: $24.5\% \rightarrow 24.1\%$) and mitigate false negatives ($\text{oLRP}_{\text{FN}}$: $39.5\% \rightarrow 38.6\%$). For a more comprehensive understanding of our findings, please refer to the experiment section.

### 3.3 Rank-oriented Matching Cost and Loss: boost the AP under high IoU thresholds

The conventional DETR and its derivatives specify the Hungarian matching cost function $\mathcal{L}_{\text{Hungarian}}$ and the training loss function $\mathcal{L}$ using the identical manner, as shown below:

$$-\lambda_1 \text{GIoU}(\hat{\mathbf{b}}, \mathbf{b}) + \lambda_2 \ell_1(\hat{\mathbf{b}}, \mathbf{b}) + \lambda_3 \text{FL}(\hat{\mathbf{p}}[c]), \tag{5}$$

where we use $\hat{\mathbf{b}}$, $\hat{\mathbf{p}}$ (or $\mathbf{b}$, $c$) to represent the predicted (or ground-truth) bounding box and classification score respectively. $c$ corresponds to the ground-truth semantic category of $\mathbf{b}$. In addition, we use the notation $\text{FL}(\cdot)$ to represent the semantic classification focal loss [41]. We propose to enhance the rank-oriented design by introducing the GIoU-aware classification head (supervised by the proposed GIoU-aware classification loss) and the high-order matching cost function scheme as follows.

**GIoU-aware Classification Loss.** Instead of applying the binary target to supervise the classification head, we propose to use the normalized GIoU scores to supervise the classification prediction:

$$\text{FL}^{\text{GIoU}}(\hat{\mathbf{p}}[c]) = -|t - \hat{\mathbf{p}}[c]|^\gamma \cdot [t \cdot \log(\hat{\mathbf{p}}[c]) + (1 - t) \cdot \log(1 - \hat{\mathbf{p}}[c])], \tag{6}$$

where we let $t = (\text{GIoU}(\hat{\mathbf{b}}, \mathbf{b}) + 1)/2$, and we denote the GIoU-aware classification loss as $\text{FL}^{\text{GIoU}}(\cdot)$. To incorporate both semantic classification and localization accuracy, we modify the original classification head to a GIoU-aware classification head supervised by the above loss function. When $t = 1$, $\text{FL}^{\text{GIoU}}(\cdot)$ simplifies to the original focal loss, denoted as $\text{FL}(\cdot)$. Furthermore, we compare our results with the varifocal loss [76], and we provide its mathematical formulation below:

$$\text{VFL}(\hat{\mathbf{p}}[c]) = -t \cdot [t \cdot \log(\hat{\mathbf{p}}[c]) + (1 - t) \cdot \log(1 - \hat{\mathbf{p}}[c])]. \tag{7}$$

**High-order Matching Cost.** In contrast to utilizing the Hungarian algorithm with a matching cost defined as the weighted sum of classification cost, $\ell_1$ loss, and GIoU loss, we propose employing a high-order matching cost function that captures a more intricate combination of the classification score and the IoU scores:

$$\mathcal{L}_{\text{Hungarian}}^{\text{high-order}} = \hat{\mathbf{p}}[c] \cdot \text{IoU}^\alpha, \tag{8}$$

where IoU represents the intersection over union scores between the predicted bounding box and the ground truth one. We adopt a larger value of $\alpha$ (e.g., >2) to prioritize the importance of localization accuracy, thereby promoting more accurate bounding box predictions and downgrading the inaccurate ones. It is worth noting that we use a high-order matching cost from the middle training stage, as most predictions exhibit poor localization quality during the early training stage.

**Analysis.** The rank-oriented loss function and matching cost are designed to enhance object detection performance at high IoU thresholds. The GIoU-aware classification loss facilitates the ranking

procedure by endowing the classification score with GIoU-awareness, resulting in more accurate ranking in the query ranking layer. Meanwhile, the high-order matching cost selects the queries with both high classification confidence and superior IoU scores as positive samples, effectively suppressing challenging negative predictions with high classification scores but low localization IoU scores. This is achieved by magnifying the advantage of a more accurate localization score using $\gamma^{\alpha}$, where $\gamma$ is the ratio of accurate localization score to less accurate score. Empirical results show significant boosts in $AP_{75}$ with GIoU-aware classification loss ($52.9\% \rightarrow 54.1\%$) or high-order matching cost design ($52.9\% \rightarrow 54.0\%$). Detailed comparisons are provided in the experiment section.

### 3.4 Discussion

The concept of GIoU-aware classification loss has been explored in several prior works [31, 37, 76, 10] predating the era of DETR. These works aimed to address the discrepancy between classification scores and localization accuracy. In line with recent concurrent research [1, 45], our method shares the same insight in designing rank-oriented matching cost and loss functions. However, our approach distinguishes itself by emphasizing the ranking aspect and introducing an additional rank-oriented architecture design, which includes the rank-adaptive classification head and query rank layer. Furthermore, our empirical results demonstrate the complementarity between the rank-oriented architecture design and the rank-oriented matching cost and loss design.

## 4 Experiment

### 4.1 Experiment Setting

We perform object detection experiments using the COCO object detection benchmark [42] with detrex [54] toolbox. Our model is trained on the `train` set and evaluated on the `val` set. We adhere to the same experimental setup as the original papers for H-DETR [30] and DINO-DETR [75].

### 4.2 Main Results

**Comparison with competing methods.** Table 1 compares Rank-DETR with other high-performing DETR-based methods on the COCO object detection `val` dataset. The evaluation demonstrates that Rank-DETR achieves remarkable results, attaining an AP score of $50.2\%$ with only 12 training epochs. This performance surpasses H-DETR [30] by $+1.5\%$ and outperforms the recent state-of-the-art method DINO-DETR [75] by $+1.2\%$ AP. Notably, we observe significant improvements in $AP_{75}$, highlighting the advantage of our approach at higher IoU thresholds.

| Method | Backbone | #query | #epochs | AP | $AP_{50}$ | $AP_{75}$ | $AP_S$ | $AP_M$ | $AP_L$ |
|---|---|---|---|---|---|---|---|---|---|
| Deformable-DETR [78] | R50 | 300 | 50 | 46.9 | 65.6 | 51.0 | 29.6 | 50.1 | 61.6 |
| DN-DETR [34] | R50 | 300 | 12 | 43.4 | 61.9 | 47.2 | 24.8 | 46.8 | 59.4 |
| DINO-DETR [75] | R50 | 900 | 12 | 49.0 | 66.6 | 53.5 | 32.0 | 52.3 | 63.0 |
| H-DETR [30] | R50 | 300 | 12 | 48.7 | 66.4 | 52.9 | 31.2 | 51.5 | 63.5 |
| Rank-DETR | R50 | 300 | 12 | 50.2 | 67.7 | 55.0 | 34.1 | 53.6 | 64.0 |

Table 1: Comparison with previous highly performant DETR-based detectors on COCO `val2017` with R50.

**Improving H-DETR [30].** Table 2 presents a detailed comparison of our proposed approach with the highly competitive H-DETR [30]. The experimental evaluation demonstrates consistent enhancements in object detection performance across various backbone networks and training schedules. For example, under the 12 epochs training schedule, our method achieves superior AP scores of $50.2\%$, $52.7\%$, and $57.3\%$ with ResNet-50, Swin-Tiny, and Swin-Large backbone networks, respectively. These results surpass the baseline methods by $+1.5\%$, $+2.1\%$, and $+1.4\%$, respectively. Extending the training schedule to 36 epochs consistently improves the AP scores, resulting in $+1.2\%$ for ResNet-50, $+1.5\%$ for Swin-Tiny, and $+1.1\%$ for Swin-Large. The AP improvement in performance is more significant under high IoU thresholds, which outperform the baseline by $+2.1\%$, $+2.7\%$, and $+1.9\%$ in $AP_{75}$ with ResNet-50, Swin-Tiny, and Swin-Large, respectively. These findings validate our proposed mechanism's consistent and substantial performance improvements across diverse settings and with different backbone networks, especially under high IoU thresholds. We also validate our performance gain by providing the PR curves under different IoU thresholds in Figure 3a.

| Method | Backbone | #epochs | AP | $AP_{50}$ | $AP_{75}$ | $AP_S$ | $AP_M$ | $AP_L$ |
|---|---|---|---|---|---|---|---|---|
| H-Deformable-DETR | R50 | 12 | 48.7 | 66.4 | 52.9 | 31.2 | 51.5 | 63.5 |
| Ours | R50 | 12 | 50.2 | 67.7 | 55.0 | 34.1 | 53.6 | 64.0 |
| H-Deformable-DETR | Swin-T | 12 | 50.6 | 68.9 | 55.1 | 33.4 | 53.7 | 65.9 |
| Ours | Swin-T | 12 | 52.7 | 70.6 | 57.8 | 35.3 | 55.8 | 67.5 |
| H-Deformable-DETR | Swin-L | 12 | 55.9 | 75.2 | 61.0 | 39.1 | 59.9 | 72.2 |
| Ours | Swin-L | 12 | 57.3 | 75.9 | 62.9 | 40.8 | 61.3 | 73.2 |
| H-Deformable-DETR | R50 | 36 | 50.0 | 68.3 | 54.4 | 32.9 | 52.7 | 65.3 |
| Ours | R50 | 36 | 51.2 | 68.9 | 56.2 | 34.5 | 54.9 | 64.9 |
| H-Deformable-DETR | Swin-T | 36 | 53.2 | 71.5 | 58.2 | 35.9 | 56.4 | 68.2 |
| Ours | Swin-T | 36 | 54.7 | 72.5 | 60.0 | 37.7 | 58.5 | 69.5 |
| H-Deformable-DETR | Swin-L | 36 | 57.1 | 76.2 | 62.5 | 39.7 | 61.4 | 73.4 |
| Ours | Swin-L | 36 | 58.2 | 76.7 | 63.9 | 42.4 | 62.2 | 73.6 |

Table 2: Improving object detection results based on H-DETR.

| Method | Backbone | #epochs | AP | $AP_{50}$ | $AP_{75}$ | $AP_S$ | $AP_M$ | $AP_L$ |
|---|---|---|---|---|---|---|---|---|
| DINO-DETR | R50 | 12 | 49.0 | 66.6 | 53.5 | 32.0 | 52.3 | 63.0 |
| Ours | R50 | 12 | 50.4 | 67.9 | 55.2 | 33.6 | 53.8 | 64.2 |
| DINO-DETR | Swin-L | 12 | 56.8 | 75.4 | 62.3 | 41.1 | 60.6 | 73.5 |
| Ours | Swin-L | 12 | 57.6 | 76.0 | 63.4 | 41.6 | 61.4 | 73.8 |

Table 3: Improving object detection results based on DINO-DETR.

**Improving DINO-DETR [75].** Table 3 shows the results of applying our approach to improve the DINO-DETR [75]. Notably, our method demonstrates an increase of +1.4% with the ResNet-50 backbone and +0.8% with the Swin-Large backbone. Under the higher IoU setting, our method further obtains +1.8% $AP_{75}$ improvement with ResNet-50 and +1.1% with Swin-Large. These results provide evidence for the generalization ability of our approach across different DETR-based models.

## 4.3 Ablation Study and Analysis

We conduct a systematic analysis to assess each proposed component's influence within our method. We followed a step-by-step approach, progressively adding modules on top of the baseline (Table 4a), incorporating each module into the baseline (Table 4b), and subsequently removing each module from our method (Table 4c). This procedure allowed us to understand the contribution of each individual component to the final performance. Furthermore, we conducted statistical and qualitative analyses to comprehensively assess the functionality of each component. We mark the best-performing numbers with ▢ colored regions in each table along each column.

**Effect of Adding Each Component Progressively.** We choose H-DETR with ResNet-50 backbone as our baseline method. By progressively adding the proposed mechanisms on top of the baseline model, it is observed that the performance is steadily increasing, and the best performance is achieved by using all the proposed components (Table 4a). It is also observed that the lowest false negative rate ($oLRP_{FN}$) is achieved when only using the rank-oriented architecture designs.

**Rank-adaptive Classification Head (RCH).** Tables 4b and 4c demonstrate that rank-adaptive classification head can slightly improve the AP (+0.2% when adding RCH to the H-DETR baseline, comparing `row1` and `row2` in Table 4b; +0.4% when adding RCH to complete our methods, comparing `row1` and `row2` in Table 4c). Furthermore, RCH improves $AP_{75}$ more than AP.

**Query Ranking Layer (QRL).** The proposed QRL mechanism effectively integrates ranking information into the DETR architecture, compensating for the absence of sequential handling of queries in attention layers. The detector's performance is also consistently improved by utilizing QRL (+0.3% when adding QRL to the H-DETR baseline; +0.7% to complete our method). We further compute the cumulative probability distribution of classification scores of positive and negative queries. QRL yields enhanced classification confidence for matched positive queries (Figure 3b),

| RCH | QRL | GCL | HMC | AP | AP$_{50}$ | AP$_{75}$ | AP$_S$ | AP$_M$ | AP$_L$ | AR$_1$ | AR$_{10}$ | AR$_{100}$ | AR$_S$ | AR$_M$ | AR$_L$ | oLRP | oLRP$_{Loc}$ | oLRP$_{FP}$ | oLRP$_{FN}$ |
|---|---|---|---|---|---|---|---|---|---|---|---|---|---|---|---|---|---|---|---|
| ✗ | ✗ | ✗ | ✗ | 48.7 | 66.4 | 52.9 | 31.2 | 51.5 | 63.5 | 37.2 | 63.4 | 68.4 | 49.7 | 72.5 | 85.9 | 61.0 | 13.3 | 24.5 | 39.5 |
| ✓ | ✗ | ✗ | ✗ | 48.9 | 66.9 | 53.3 | 31.2 | 52.4 | 63.7 | 37.5 | 64.4 | 71.2 | 53.5 | 75.5 | 87.1 | 61.2 | 13.3 | 24.0 | 39.2 |
| ✓ | ✓ | ✗ | ✗ | 49.3 | 67.3 | 53.7 | 32.4 | 52.2 | 63.4 | 37.8 | 65.0 | 71.7 | 54.6 | 75.7 | 88.4 | 60.8 | 13.3 | 24.1 | 38.6 |
| ✓ | ✓ | ✓ | ✗ | 49.8 | 67.5 | 54.3 | 33.3 | 53.4 | 63.7 | 37.9 | 65.1 | 71.8 | 54.8 | 76.1 | 87.8 | 60.7 | 13.0 | 23.5 | 39.3 |
| ✓ | ✓ | ✓ | ✓ | 50.2 | 67.7 | 55.0 | 34.1 | 53.6 | 64.0 | 38.1 | 64.9 | 71.6 | 56.5 | 75.8 | 86.4 | 60.4 | 12.9 | 22.4 | 39.3 |

(a) Effect of gradually adding modules on the baseline.

| RCH | QRL | GCL | HMC | AP | AP$_{50}$ | AP$_{75}$ | AP$_S$ | AP$_M$ | AP$_L$ | AR$_1$ | AR$_{10}$ | AR$_{100}$ | AR$_S$ | AR$_M$ | AR$_L$ | oLRP | oLRP$_{Loc}$ | oLRP$_{FP}$ | oLRP$_{FN}$ |
|---|---|---|---|---|---|---|---|---|---|---|---|---|---|---|---|---|---|---|---|
| ✗ | ✗ | ✗ | ✗ | 48.7 | 66.4 | 52.9 | 31.2 | 51.5 | 63.5 | 37.2 | 63.4 | 68.4 | 49.7 | 72.5 | 85.9 | 61.0 | 13.3 | 24.5 | 39.5 |
| ✓ | ✗ | ✗ | ✗ | 48.9 | 66.9 | 53.3 | 31.2 | 52.4 | 63.7 | 37.5 | 64.4 | 71.2 | 53.5 | 75.5 | 87.1 | 61.2 | 13.3 | 24.0 | 39.2 |
| ✗ | ✓ | ✗ | ✗ | 49.0 | 67.2 | 53.2 | 32.3 | 51.9 | 63.5 | 37.9 | 64.8 | 71.5 | 55.1 | 75.6 | 87.5 | 61.1 | 13.4 | 23.8 | 39.2 |
| ✗ | ✗ | ✓ | ✗ | 49.4 | 67.0 | 54.1 | 32.0 | 52.8 | 64.0 | 37.9 | 64.9 | 71.7 | 55.4 | 75.7 | 88.5 | 60.7 | 12.9 | 22.5 | 39.8 |
| ✗ | ✗ | ✗ | ✓ | 49.3 | 67.3 | 54.0 | 31.8 | 52.4 | 63.4 | 37.7 | 64.4 | 71.1 | 53.7 | 74.9 | 85.8 | 61.1 | 13.2 | 23.8 | 39.2 |

(b) Effect of incorporating each module on the baseline.

| RCH | QRL | GCL | HMC | AP | AP$_{50}$ | AP$_{75}$ | AP$_S$ | AP$_M$ | AP$_L$ | AR$_1$ | AR$_{10}$ | AR$_{100}$ | AR$_S$ | AR$_M$ | AR$_L$ | oLRP | oLRP$_{Loc}$ | oLRP$_{FP}$ | oLRP$_{FN}$ |
|---|---|---|---|---|---|---|---|---|---|---|---|---|---|---|---|---|---|---|---|
| ✓ | ✓ | ✓ | ✓ | 50.2 | 67.7 | 55.0 | 34.1 | 53.6 | 64.0 | 38.1 | 64.9 | 71.6 | 56.5 | 75.8 | 86.4 | 60.4 | 12.9 | 22.4 | 39.3 |
| ✗ | ✓ | ✓ | ✓ | 49.8 | 67.3 | 54.5 | 33.5 | 53.4 | 63.6 | 38.1 | 64.9 | 71.5 | 55.9 | 75.3 | 86.6 | 60.5 | 12.9 | 22.9 | 39.6 |
| ✓ | ✗ | ✓ | ✓ | 49.5 | 67.4 | 54.2 | 33.1 | 52.8 | 63.5 | 38.0 | 64.7 | 71.6 | 55.8 | 75.4 | 85.9 | 60.9 | 12.9 | 24.6 | 39.3 |
| ✓ | ✓ | ✗ | ✓ | 49.5 | 67.6 | 54.1 | 32.4 | 52.6 | 64.3 | 37.9 | 64.3 | 71.0 | 54.4 | 75.1 | 85.4 | 60.8 | 13.4 | 23.7 | 38.7 |
| ✓ | ✓ | ✓ | ✗ | 49.8 | 67.5 | 54.3 | 33.3 | 53.4 | 63.7 | 37.9 | 65.1 | 71.8 | 54.8 | 76.1 | 87.8 | 60.7 | 13.0 | 23.5 | 39.3 |

(c) Effect of removing each module in our method.

Table 4: Ablation experiments based on H-DETR + R50. RCH: rank-adaptive classification head. QRL: query rank layer. GCL: GIoU-aware classification loss. HMC: high-order matching cost.

| Classification loss target | AP | AP$_{50}$ | AP$_{75}$ | AP$_S$ | AP$_M$ | AP$_L$ | AR$_1$ | AR$_{10}$ | AR$_{100}$ | AR$_S$ | AR$_M$ | AR$_L$ | oLRP | oLRP$_{Loc}$ | oLRP$_{FP}$ | oLRP$_{FN}$ |
|---|---|---|---|---|---|---|---|---|---|---|---|---|---|---|---|---|
| $t = \text{IoU}(\hat{\mathbf{b}}, \mathbf{b})^{0.5}$ | 50.1 | 67.6 | 54.7 | 32.6 | 53.4 | 64.5 | 38.2 | 64.9 | 71.5 | 55.5 | 75.4 | 86.4 | 60.4 | 12.9 | 22.5 | 39.4 |
| $t = \text{IoU}(\hat{\mathbf{b}}, \mathbf{b})^{1}$ | 50.0 | 67.3 | 54.7 | 34.0 | 53.6 | 64.8 | 38.1 | 65.2 | 71.7 | 56.7 | 75.5 | 85.8 | 60.6 | 12.9 | 22.9 | 39.6 |
| $t = \text{IoU}(\hat{\mathbf{b}}, \mathbf{b})^{2}$ | 49.5 | 66.0 | 54.0 | 32.5 | 53.3 | 64.1 | 37.8 | 64.2 | 70.9 | 53.8 | 75.3 | 86.1 | 61.0 | 12.3 | 23.1 | 41.0 |
| $t = [(\text{GIoU}(\hat{\mathbf{b}}, \mathbf{b}) + 1)/2]^{0.5}$ | 49.9 | 67.9 | 54.3 | 32.8 | 53.2 | 64.3 | 37.9 | 64.9 | 71.6 | 55.8 | 75.5 | 86.1 | 60.6 | 13.2 | 23.3 | 38.7 |
| $t = [(\text{GIoU}(\hat{\mathbf{b}}, \mathbf{b}) + 1)/2]^{1}$ | 50.2 | 67.7 | 55.0 | 34.1 | 53.6 | 64.0 | 38.1 | 64.9 | 71.6 | 56.5 | 75.8 | 86.4 | 60.4 | 12.9 | 22.4 | 39.3 |
| $t = [(\text{GIoU}(\hat{\mathbf{b}}, \mathbf{b}) + 1)/2]^{2}$ | 50.1 | 67.4 | 54.9 | 33.3 | 53.5 | 64.3 | 38.1 | 64.9 | 71.5 | 56.3 | 75.3 | 85.5 | 60.4 | 12.9 | 22.9 | 39.3 |

(a) Effect of the GIoU-aware classification loss target formulation.

| Matching cost | AP | AP$_{50}$ | AP$_{75}$ | AP$_S$ | AP$_M$ | AP$_L$ | AR$_1$ | AR$_{10}$ | AR$_{100}$ | AR$_S$ | AR$_M$ | AR$_L$ | oLRP | oLRP$_{Loc}$ | oLRP$_{FP}$ | oLRP$_{FN}$ |
|---|---|---|---|---|---|---|---|---|---|---|---|---|---|---|---|---|
| $\hat{\mathbf{p}}[c] \cdot \text{IoU}^1$ | 48.3 | 67.9 | 51.8 | 31.4 | 51.3 | 62.3 | 37.8 | 62.9 | 67.7 | 52.8 | 70.9 | 82.6 | 60.4 | 13.5 | 22.8 | 38.4 |
| $\hat{\mathbf{p}}[c] \cdot \text{IoU}^2$ | 49.5 | 68.3 | 53.5 | 32.5 | 52.4 | 63.9 | 38.1 | 63.9 | 69.6 | 54.5 | 73.3 | 84.8 | 60.2 | 13.3 | 23.1 | 37.8 |
| $\hat{\mathbf{p}}[c] \cdot \text{IoU}^3$ | 50.0 | 68.1 | 54.2 | 32.9 | 53.2 | 64.4 | 37.9 | 64.7 | 70.9 | 55.3 | 74.9 | 85.9 | 60.4 | 12.9 | 22.9 | 39.1 |
| $\hat{\mathbf{p}}[c] \cdot \text{IoU}^4$ | 50.2 | 67.7 | 55.0 | 34.1 | 53.6 | 64.0 | 38.1 | 64.9 | 71.6 | 56.5 | 75.8 | 86.4 | 60.4 | 12.9 | 22.4 | 39.3 |
| $\hat{\mathbf{p}}[c] \cdot \text{IoU}^5$ | 50.0 | 67.1 | 54.9 | 32.4 | 53.5 | 64.5 | 38.1 | 65.0 | 71.8 | 55.1 | 75.9 | 86.5 | 60.7 | 12.8 | 23.4 | 39.4 |
| $\hat{\mathbf{p}}[c] \cdot \text{IoU}^6$ | 50.0 | 66.6 | 54.9 | 33.9 | 53.3 | 64.5 | 38.0 | 65.4 | 72.5 | 56.3 | 76.5 | 87.7 | 61.0 | 12.8 | 23.0 | 40.3 |
| $\hat{\mathbf{p}}[c] \cdot (\frac{\text{GIoU}+1}{2})^1$ | 46.6 | 65.9 | 49.9 | 31.0 | 49.8 | 59.9 | 37.2 | 60.3 | 63.4 | 47.9 | 66.9 | 77.5 | 60.6 | 13.8 | 22.9 | 38.2 |
| $\hat{\mathbf{p}}[c] \cdot (\frac{\text{GIoU}+1}{2})^2$ | 47.8 | 67.2 | 51.0 | 31.6 | 50.8 | 61.8 | 37.6 | 61.9 | 65.9 | 50.8 | 68.6 | 80.4 | 60.4 | 13.5 | 22.5 | 38.2 |
| $\hat{\mathbf{p}}[c] \cdot (\frac{\text{GIoU}+1}{2})^3$ | 48.0 | 67.6 | 51.3 | 31.1 | 51.3 | 62.2 | 37.6 | 61.8 | 65.8 | 50.2 | 69.3 | 79.8 | 59.9 | 13.4 | 21.9 | 37.9 |
| $\hat{\mathbf{p}}[c] \cdot (\frac{\text{GIoU}+1}{2})^4$ | 49.1 | 68.2 | 53.0 | 32.2 | 52.1 | 63.5 | 37.7 | 63.9 | 69.2 | 55.0 | 72.2 | 83.6 | 60.1 | 13.3 | 22.9 | 37.7 |
| $\hat{\mathbf{p}}[c] \cdot (\frac{\text{GIoU}+1}{2})^5$ | 49.4 | 68.5 | 53.0 | 32.6 | 52.7 | 63.7 | 38.2 | 63.9 | 69.1 | 54.0 | 72.7 | 83.8 | 60.0 | 13.2 | 21.6 | 38.4 |
| $\hat{\mathbf{p}}[c] \cdot (\frac{\text{GIoU}+1}{2})^6$ | 49.8 | 68.4 | 54.1 | 32.9 | 53.4 | 63.9 | 38.2 | 64.8 | 70.6 | 55.3 | 74.8 | 85.5 | 60.1 | 13.1 | 23.7 | 37.7 |
| $\hat{\mathbf{p}}[c] \cdot (\frac{\text{GIoU}+1}{2})^7$ | 49.9 | 67.9 | 54.2 | 32.6 | 53.0 | 64.6 | 38.1 | 64.5 | 70.5 | 55.0 | 74.5 | 85.7 | 60.3 | 13.0 | 23.0 | 38.7 |
| $\hat{\mathbf{p}}[c] \cdot (\frac{\text{GIoU}+1}{2})^8$ | 50.0 | 67.8 | 54.6 | 33.9 | 53.6 | 64.7 | 37.9 | 65.0 | 71.2 | 55.5 | 75.2 | 86.5 | 60.4 | 12.8 | 23.8 | 38.8 |
| $\hat{\mathbf{p}}[c] \cdot (\frac{\text{GIoU}+1}{2})^9$ | 49.8 | 67.3 | 54.4 | 33.3 | 53.2 | 63.7 | 38.1 | 65.1 | 71.7 | 56.4 | 75.7 | 87.1 | 60.5 | 12.9 | 23.9 | 39.0 |

(b) Effect of the matching cost formulation.

Table 5: The influence of classification loss target and matching cost function choice.

while the classification confidence for unmatched queries is effectively suppressed (Figure 3c), thereby ranking true predictions higher than potential false predictions. This phenomenon is further apparent from the oLRP$_{FP}$ results showcased in Table 4b, which is reduced from $24.5\%$ to $23.8\%$ by using QRL, reducing the false positive rates. These results are in accordance with our design intent.

**GIoU-aware Classification Loss (GCL).** Table 4 also shows the effectiveness of the proposed GIoU-aware classification loss, which gains $0.7\%$ mAP over the vanilla baseline by comparing `row1` and `row4` in Table 4b and $0.7\%$ mAP increase by comparing `row1` and `row4` in Table 4c. We

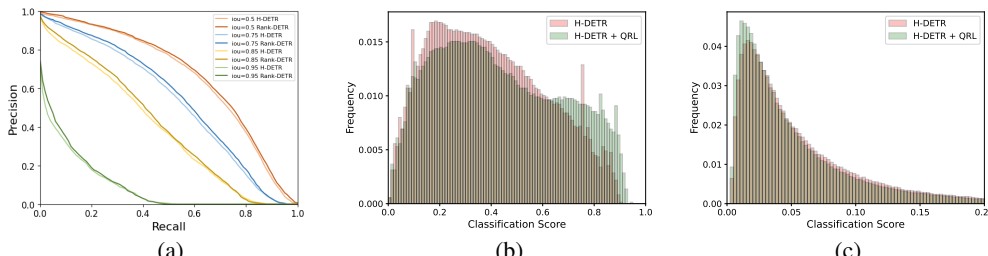

(a)  (b)  (c)

Figure 3: (a) Comparing the PR-curves between baseline and our method under different IoU thresholds. (b) Density distribution of the classification scores on the matched queries with or without QRL. (c) Density distribution of the classification scores on the unmatched queries with or without QRL.

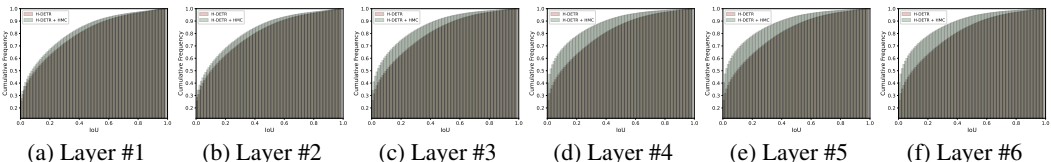

(a) Layer #1    (b) Layer #2    (c) Layer #3    (d) Layer #4    (e) Layer #5    (f) Layer #6

Figure 4: Cumulative distribution of the IoU scores on the unmatched queries with or without HMC.

also ablate the formulation of the learning target $t$ in Eq. (6) in Table 5a. The results show that the performance of adopting IoU (and its exponent) as the optimization target is inferior to use $(\text{GIoU} + 1)/2$ because GIoU can better model the relationship of two non-overlapped boxes. We use $(\text{GIoU} + 1)/2$ rather than GIoU because $-1 < \text{GIoU} \leq 1$ and $0 < (\text{GIoU} + 1)/2 \leq 1$.

**High-order Matching Cost (HMC).** We also show how the high-order matching cost affects the overall performance in Table 4. The HMC can also significantly improve the overall performance of the object detector ($+0.6\%$ when comparing `row1` and `row5` in Table 4b, $+0.4\%$ when comparing `row1` and `row5` in Table 4c). We further ablate the formulation of the matching cost. As illustrated in Table 5b, a high-order exponent IoU can consistently improve the performance and achieve a peak when the power is $4$. Using a high-order exponent can suppress the importance of the predicted boxes with low IoU. We can also observe the same trend from Table 5b when replacing IoU with $(\text{GIoU}+1)/2$, but the latter practice has a slightly inferior performance.

**HMC Suppresses the Overlap between Negative Query and Ground Truth**. Figure 4 illustrates the cumulative probability distribution of the IoU of unmatched queries. The IoU of each unmatched query is defined as the largest IoU between it and all ground truth boxes. As shown in Figure 4, the adoption of HMC can decrease the IoU between unmatched queries and all the ground truth bounding boxes, effectively pushing the negative queries away from the ground truth boxes. Furthermore, this phenomenon is increasingly remarkable in the latter Transformer decoder layers.

**Comparison with Varifocal loss.** In order to assess the effectiveness of the proposed GIoU-aware classification loss (GCL, Eq. (6)), we compare it with varifocal loss (VFL) [76] due to their similar mathematical formulations. Following VFL [76], we utilize the training target $t = \text{IoU}$ in Eq. (7). To simplify the comparison and focus on the impact of GCL, we conduct the evaluation without HMC (`row3` in Table 4a). Our method, leveraging the GCL, achieves a mAP of $49.8\%$ (`row4` in Table 4a), whereas VFL achieves only $49.5\%$ mAP. The primary distinctions between GCL and VFL lie in the optimization target. By employing the normalized GIoU as the training target, our approach better models the distance between two non-overlapping boxes, leading to improved performance. In addition, VFL removes scaling factors on positive samples, as they are rare compared to negatives in CNN-based detectors. However, for DETR-based detectors, where positive examples are relatively more abundant, we empirically show that retaining a scaling factor can enhance performance.

| Method | Backbone | Params(M) | FLOPs(G) | Training Cost (min) | Testing FPS (img/s) | AP |
|--------|----------|-----------|----------|---------------------|---------------------|------|
| H-DETR | R50 | 47.56 | 280.30 | 69.8 | 19.2 | 48.7 |
| Ours | R50 | 49.10 | 280.60 | 71.8 | 19.0 | 50.2 |

Table 6: Computational efficiency analysis for our method.

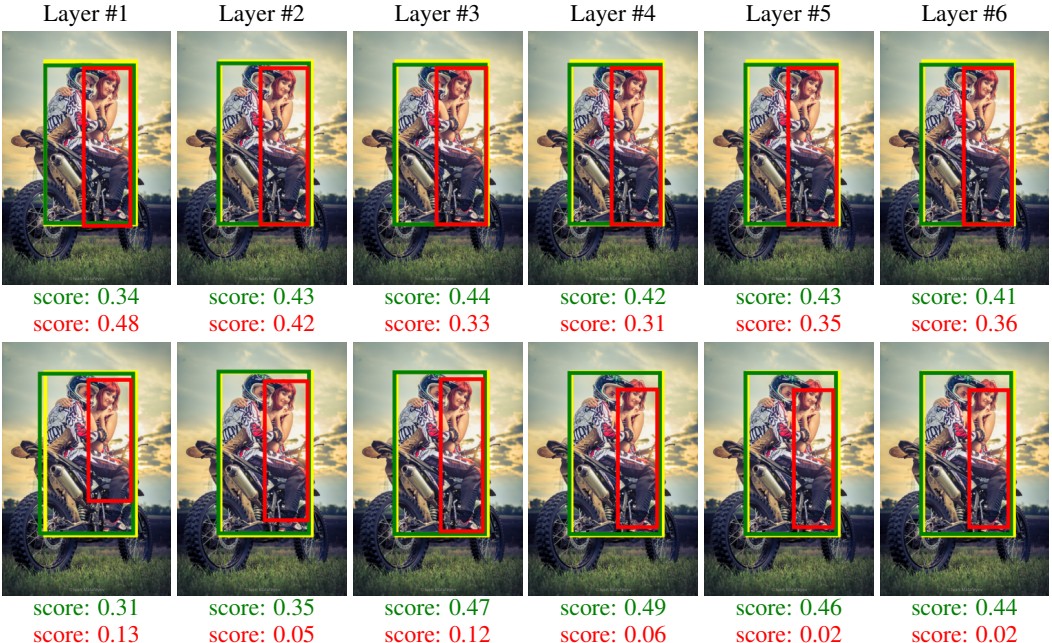

Figure 5: Illustration of positive boxes and negative box of H-DETR (`row1`) and Rank-DETR (`row2`). The green, red and yellow boxes (scores) refer to the positive, negative, and ground truth boxes (scores), respectively.

**Computational Efficiency.** In Table 6, we provide comprehensive data on the number of parameters, computational complexity measured in FLOPs, training time per epoch, inference frames per second (FPS), and Average Precision performance, for both the H-DETR baseline and our approach. These assessments were conducted on RTX 3090 GPUs, allowing us to evaluate testing and training efficiency. The results unequivocally highlight a substantial enhancement in detection performance achieved by our proposed method, with only a slight increase in FLOPs and inference latency. Considering the effectiveness and efficiency, our method has the potential to be adapted into 3D object detection [55, 29], semantic segmentation [39, 33] tasks, or other applications [23, 24, 25, 21, 22].

**Qualitative Analysis.** Figure 5 visualizes the predicted bounding boxes and their classification confidence scores of both the matched positive queries and the unmatched hard negative queries, respectively. We find that, compared to the baseline method (`row1`), the proposed approach (`row2`) effectively promotes the classification scores of positive samples, while that of hard negative queries is rapidly suppressed, progressing layer by layer. These qualitative results further illustrate how the proposed approach achieves high performance by decreasing the false positive rate.

## 5 Conclusion

This paper presents a series of simple yet effective rank-oriented designs to boost the performance of modern object detectors and result in the development of a high-quality object detector named Rank-DETR. The core insight behind the effectiveness lies in establishing a more precise ranking order of predictions, thereby ensuring robust performance under high IoU thresholds. By incorporating accurate ranking order information into the network architecture and optimization procedure, our approach demonstrates improved performance under high IoU thresholds within the DETR framework. While there remains ample scope to explore leveraging rank-oriented designs, we hope that our initial work serves as an inspiration for future efforts in building high-quality object detectors.

## Acknowledgement

This work is supported by National Key R&D Program of China under Grant 2022ZD0114900 and 2018AAA0100300, National Nature Science Foundation of China under Grant 62071013 and 61671027. We also appreciate Ding Jia, Yichao Shen, Haodi He and Yutong Lin for their insightful discussions, as well as the generous donation of computing resources by High-Flyer AI.

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
