# OpenReview forum: "Rank-DETR for High Quality Object Detection"
_NeurIPS.cc/2023/Conference — NeurIPS 2023 poster_

### Official Review · Reviewer_kDha · 2023-07-03

**Soundness:** 3 good
**Presentation:** 3 good
**Contribution:** 2 fair
**Rating:** 5
**Confidence:** 3

**Summary:**

This paper focuses on the ranking problem in object detection. Inspired by the misalignment between class and location scores in DTER models, the paper proposes to redesign the model architecture and the loss to modulate the rank information. Experiments show that , the proposed method can build upon the current strong DETR detectors such as DINO and H-DETR to further boost the performance by a notable margin (>1 mAP). The experiments also suggest the complementarity between the ranking model architecture and the ranking loss design.

**Strengths:**

+ The paper tackles an important problem in DETRs for object detection: the misalignment of the class scores and the localization scores.
+ The paper is well written and easy to follow. The paper has provided informative statistical plots. For example, the density distribution for QRL and HWC, and the illustration of box scores with and without the proposed ranking mechanism.
+ The experiments are solid and sound. The paper uses detailed ablation studies to justify the design choices of the proposed components. The method seems effective according to the experimental results, i.e., around +1% AP metric for DINO and H-DETR.


**Weaknesses:**

- The whole process seems a little engineering. E.g., the addition of learnable bias.
- The main boost originates from the GIoU-aware loss, which has been investigated in Varifocal loss, Align-DETR and Stable-DETR (I am aware that the latter two are not formally published). This somehow hurts the novelty of the paper. A thorough discussion/comparison is expected.
- The method seems complicated and includes many steps. For example the rank-driven model design seems not to invite as much performance gains as expected.


**Questions:**

- The title is too broad and has little meaning. I recommend to change a name including the technical contribution.
- The addition of the RCH module even hurts the performance, especially for AP_L, it degrades 0.6 points. Please explain this phenomenon.
- The learnable logit bias vectors S and the randomly initialized content query C are confusing to me. Are they tricks or can the author give an explanation (or illustrations with figures) for their practical functions? I think further ablations on the two learnable bias are required.
- The result for DINO, R50, 12epochs in Table 1 is different from that in Table 3. Please check.


**Limitations:**

The current solution is somewhat straightforward, and the topic can be deep investigated.

---

> ### Author Rebuttal · Authors · 2023-08-10
>
> ## To Reviewer kDha
>
> Thanks for your detailed comments. The mentioned questions are addressed as follows.
>
> ---
>
> > **The whole process seems a little engineering. E.g., the addition of learnable bias.**
>
> A: We appreciate your feedback and are grateful for the opportunity to address your concern regarding our work is engineering. We agree some of the proposed designs essentially require engineering techniques and they are important to build high quality object detectors. We also would like to further simply the engineering design in the short future.
>
> ---
>
> > **The main boost originates from the GIoU-aware loss, which has been investigated in Varifocal loss, Align-DETR and Stable-DETR (I am aware that the latter two are not formally published). This somehow hurts the novelty of the paper. A thorough discussion/comparison is expected.**
>
> A: Great point! First, we want to clarify that the other proposed components also bring significant improvements. For example, according to Table 4(a), RCH improves AP from 48.2 to 48.4 (+0.2), QRL improves AP from 48.4 to 49.1 (+0.7), GCL (GIoU-aware loss) improves AP from 49.1 to 49.6 (+0.5), and HMC improves AP from 49.6 to 50.0 (+0.4). Therefore, we can see that the proposed GIoU-aware loss contributes less than 30% of the overall gains. A thorough discussion/comparison with Align-DETR and Stable-DINO is provided in the general response.
>
> ---
>
> > **The method seems complicated and includes many steps. For example the rank-driven model design seems not to invite as much performance gains as expected.**
>
> A: First, we want to clarify that the proposed method is easy to implement. RCH and QRL can be implemented with fewer than 100 lines of code, while GCL is a simple modification of Focal Loss that changes the hard label into a soft GIoU target. HMC is a simplified version of an existing matching cost design, which reduces the conventional weighted sum of three costs to just one term.
>
> Second, although we propose four components, it is a comprehensive design that introduces ranking into DETRs. It includes both architectural design and optimization modifications.
>
> Third, the proposed components all facilitate better ranking. RCH and QRL embed ranking information in the network architecture. GCL and HMC make the ranking metric (classification score) IoU/GIoU aware and further improve the quality of ranking.
>
> ---
>
> > **The title is too broad and has little meaning. I recommend to change a name including the technical contribution.**
>
> A: Good point! We appreciate your valuable suggestions on including the technical contribution in the title. The technical contribution of this work lies in introducing rank-oriented architecture designs and rank-oriented optimization designs (training loss and matching cost). Therefore, we propose a possible candidate: “Rank-DETR: Improving DETR with Rank-Oriented Architecture and Loss Designs”. We would appreciate any further valuable comments you may have.
>
> ---
>
> > **The addition of the RCH module even hurts the performance, especially for AP_L, it degrades 0.6 points. Please explain this phenomenon.**
>
> A: In our observation, large objects tend to have a high classification score in DETRs, while the small ones do not. The proposed RCH tends to enhance the classification score of small objects. As a result, the average precision (AP) for small objects is improved, while AP_L is affected negatively. Overall, RCH still increases the overall performance in terms of the mean average precision (mAP) metric.
>
> ---
>
> > **The learnable logit bias vectors S and the randomly initialized content query C are confusing to me. Are they tricks or can the author give an explanation (or illustrations with figures) for their practical functions? I think further ablations on the two learnable bias are required.**
>
> A: The function of rank-aware static content query C has been analyzed in Fig. 3(b) and (c). Fig. 3(b) shows that by embedding C in the sorted content queries O, the classification scores of positive queries (matched in the Hungarian matching process) are enhanced. Fig. 3(c) illustrates that the score of negative queries (unmatched ones) is suppressed. Therefore, QRL ensures lower false positives (FP) and false negatives (FN).
> We further visualize the learnable logit bias vectors S for each decoder layer in the **attached PDF in the general response** (Fig. 6 and Fig. 7). For each decoder layer, the logit bias is a tensor with a shape of [num\_queries, num\_classes]. We use the index of queries (which has been sorted) as the x-axis and the processed value of logits (mean for Fig. 6, max for Fig. 7) for the y-axis. It can be observed that the learned logit bias further enhances the higher-ranked logits and suppresses the lower-ranked ones.
>
> ---
>
> > **The result for DINO, R50, 12epochs in Table 1 is different from that in Table 3. Please check.**
>
> A: Good point! The results in Table 1 are the numbers reported in the original paper. The results in Table 3 were reproduced by our server for a fair comparison. Our reproduced performance of DINO-DETR is slightly worse than the reported numbers in the original paper. We appreciate your suggestion and would like to explicitly state that the numbers in Table 3 were reproduced by us based on your valuable comments.

---

> > ### Comment · Reviewer_kDha · 2023-08-18
> > **Thanks the authors for the rebuttal**
> >
> > I have read the rebuttal and other reviews. The rebuttal has addressed my concerns sufficiently. I would like to keey my original rating as "borderline accept".

---

### Official Review · Reviewer_vAjC · 2023-07-03

**Soundness:** 3 good
**Presentation:** 3 good
**Contribution:** 3 good
**Rating:** 7
**Confidence:** 5

**Summary:**

In this paper, the authors study the problem of object detection. To be specific, they introduce rank-awareness into transformer-based detectors both at the architecture-level and the loss/cost-level.

After the rebuttal:

The authors have addressed my concerns about the comparison with other ranking-based solutions. Therefore, I recommend the paper to be accepted.

**Strengths:**

1. Ranking is an important problem in object detection.

2. The paper introduces and incorporates many interesting ways of integrating ranking into transformer-based detectors.

3. Strong results compared to existing methods.

**Weaknesses:**

1. The paper is missing citations and comparison to significant ranking-based object detectors. To name a few, AP Loss, aLRP Loss, Rank & Sort Loss, Correlation Loss.

2. "Rank-adaptive Classification Head" => What does it have to do with ranking? This module just adds a learnable bias to classification scores, which do not depend on negatives or positives or localization qualities.

3. Some aspects require clarifications:

3.1. Eq 2: How does updating the positional encodings based on the sorted boxes disrupt positional information of the boxes?

3.2. Eq 2 & 3: Do you not need Sort() to be differentiable to pass gradients through?

3.3. "High-order Matching Cost" => Why high-order? What is high and what is order in this context?

4. Improvement over Varifocal loss (0.3% mAP) is insignificant. It is not clear what was missing in existing ranking-based loss functions.

5. Presumably there was not sufficient time to compare against [1, 21]. If you have performed a comparison in the meantime, can you please share the results?

6. It is a pity that there are no results on the LVIS dataset.

**Questions:**

Please see above.

**Limitations:**

None.

---

> ### Author Rebuttal · Authors · 2023-08-10
>
> ## To Reviewer vAjC
> ---
>
> > **Missing citations and comparison to significant ranking-based object detectors.**
>
> A: Thanks for sharing so many valuable ranking-based object detectors which surely will help us to improve our work! We would like include the missing citations and the following comparisons in the revision.
>
> 👉 First, we discuss their mathmatical formulation differences as follows:
>
> - AP Loss: $L_{AP} = \frac{1}{|P|} \sum_{i\in P} \sum_{j\in N}L_{ij}$, where the AP loss aims to address the extreme foreground background class imbalance issue.
>
> - aLRP Loss: $L^{aLRP}=\frac{1}{|P|}\sum_{i\in P}\ell^{\mathrm{LRP}}(i)$, where the aLRP loss is the  first ranking based loss function for both classification and localisation tasks.
>
> - Rank and Sort Loss: $L_{RS} := \frac{1}{|P|} \sum_{i\in \mathcal{P}} (\ell_{\mathrm{RS}}(i) - \ell_{\mathrm{RS}}^*(i))$, where the Rank and Sort Loss aims to rank each positive above all negatives as well as to sort positives among themselves with respect to their localisation qualities.
>
> - Correlation Loss: $L_{corr} =1 − \rho(IoU,s)$, where the Correlation Loss is a simple plug-in loss function to improve correlation of classification and localization tasks.
>
> According to the above formulations, we can see that AP Loss, aLRP Loss, and Rank Sort Loss mainly constrain the classification and localization predictions between pair-wise samples (positive sample pairs or positive-negative sample pairs), using error-driven update. However, our loss function use plain backpropagation to align each classification score with its target, but not between each pair of classification scores. In addition, correlation loss is a plug-in component used together with the classification loss, while our loss function replaces the original classification loss.
>
>
> 👉 Second, we also attempt to integrate these methods with the H-DETR method and report the initial comparison results as follows:
>
> | method |  mAP | AP50 |  AP75|
> | ---- | --- | ---- | ----- |
> | H-DETR (reproduced baseline) | 48.2 |  66.4 | 52.9 |
> | H-DETR + Correlation Loss | 48.9 (+0.7) |  65.7 | 53.3 |
> | H-DETR + Ours |  49.2 (+1.0) | 66.9 | 53.7 |
>
> Also, other methods (AP, aLRP, RankSort) result in slight mAP degradations, and our method significantly outperforms these ranking losses based on H-DETR. This results from the choice of the **matching strategy** between the detectors used in these papers and DETR. We would like to include above discussions in the final revision and welcome any further suggestions.
>
>
> ---
> > **Ranking in RCH.**
>
> A: The input $\mathbf{o}^l_i$ for Equation $\mathbf{t}^l_i=\operatorname{MLP}(\mathbf{o}^l_i)$ is
> computed based on the sorted content query $\overline{\mathcal{Q}}_c^{l-1}$ and sorted position query $\overline{\mathcal{Q}}_p^{l-1}$ from the previous decoder layer.
> Therefore, each element of the learnable bias corresponds to a specific ranking position according to the descending order of the classification scores predicted by the previous decoder layer.
>
>
> ---
> > **clarifications**
>
> A:
>
> 👉 Eq 2: How does updating the positional encodings based on the sorted boxes disrupt positional information of the boxes?
>
> We update the positional encodings based on the sorted boxes in order to **keep the order of the positional embedding aligned with the order of the ranked content query** as we need to send the combination of rank-aware positional query embedding and content query embedding into the subsequent Transformer decoder layers.
>
> 👉 Eq 2 & 3: Do you not need Sort() to be differentiable to pass gradients through?
>
> Yes, we implement the Sort() function in Eq 2 & 3 with the combination of two functions including torch.argsort() and torch.gather(). First, we use torch.argsort() to get the indices of the sorted probability predictions $\mathcal{P}^{l}$ and torch.argsort() is not differentiable. Second, we use torch.gather() to gather the values of $\mathcal{B}^{l}$ and $\mathcal{O}^{l}$ according to the output of torch.argsort(). The operation torch.gather() is differentiable to pass gradients through.
>
> 👉 "High-order Matching Cost" => Why high-order? What is high and what is order in this context?
>
> (i) The reason for using a high-order combination of the class scores and the IoU is:
>
> - **non-linear relationships**: using high-order combinations allows us to capture non-linear relationships between the classification scores and the IoU scores, leading to better model performance and a more accurate representation of the underlying patterns in the data.
>
> - **better representation of interaction effects**: high-order combinations can capture interaction effects between two variables more effectively than linear combinations. We can use different power terms for them to control the effect of each variable more flexibly.
>
> (ii) In this work, we use high-order to refer to non-linear combinations of the classification scores and the IoU scores, i.e., $\hat{\mathbf{p}}[l] \cdot {\text{IoU}}^{\alpha}$, as shown in Equation (7).
>
> ---
> > **Varifocal loss**
>
> A: Varifocal loss **removes the scaling factors on positive examples**. According to original paper, the reason for such a design is because **positive examples are extremely rare compared with negatives and we should keep their precious learning signals**. However, this assumption is not true for DETR-based detectors, where **the ratio of positive examples is relatively much larger** (e.g., 30-50 positive samples vs. 250-270 negative samples given 300 queries). Therefore, we empirically show that applying a scaling factor ($-|t-\hat{\mathbf{p}}[l]|^\gamma$) to positive samples can help improve the performance.
>
> ---
> > **Comparison with Aligh-DETR, Stable-DINO**
>
> A: Please refer to the general response.
>
> ---
> > **For LVIS dataset**
>
> A: We provide the results on LVIS based on Detic with Deformable-DETR + R50 (https://github.com/HDETR/H-Detic-LVIS) trained for 24 epochs:
>
> | method     | mAP |
> | -| - |
> | Detic      | 30.9 |
> | Rank Detic  | 34.1 |

---

> > ### Comment · Reviewer_vAjC · 2023-08-13
> > **Re: Rebuttal by Authors**
> >
> > Thank you for the detailed rebuttal. I am happy with the responses and the new results. Therefore, I will keep my original recommendation as Accept.

---

> > > ### Author Response · Authors · 2023-08-13
> > > **Thanks for the Response of Reviewer vAjC**
> > >
> > > We thank the reviewer for your prompt response and for keeping the positive rating.
> > >
> > > We intend to incorporate the rebuttal contents into the final revision in line with your invaluable suggestions. Your guidance has been instrumental in enhancing the quality of our work, and we truly appreciate your dedication to this process.

---

### Official Review · Reviewer_JDo7 · 2023-07-05

**Soundness:** 2 fair
**Presentation:** 2 fair
**Contribution:** 2 fair
**Rating:** 6
**Confidence:** 4

**Summary:**

This paper proposes a DETR training method named Rank DETR that integrates multiple (four) rank-oriented designs, i.e., rank-adaptive classification head (RCH), query ranking layer (QRL), GIoU-aware classification loss (GCL), and high-order matching cost (HMC). Among these four components, the former two (RCH and QRL) are relatively novel, while the latter two (GCL and HMC) are close to already-existing IoU-based loss function and matching criterion (e.g., as in Stable DINO). Experimental results show that the proposed Rank DETR improves multiple DETR baselines. However, some recent methods (e.g., Stable DINO) with fewer components achieve comparable or even higher results.

**Strengths:**

- Two (out of four) major components, i.e., rank-adaptive classification head (RCH), query ranking layer (QRL) are novel.
- Ablation experiments show that most components  bring considerable improvement (the improvement from RCH is relatively trivial) and integrating them brings further improvement.


**Weaknesses:**

- Two (out of four) major components, i.e.,  GIoU-aware classification loss (GCL), and high-order matching cost (HMC), share close insight, motivation and mechanism with recent methods, e.g., Stable DINO and Aligned DETR (though the detailed implementation is different). Moreover, the improvement brought by these two components is smaller than the similar ones in the competing methods.
- More importantly, though the proposed Rank DETR adds two more components (RCH and QRL) based on IoU-related loss and matching (GCL and HMC), the overall results of Rank DETR is still lower than the recent methods that only has IoU-related loss (and matching), i.e., Stable DINO. For example, with DINO baseline, the proposed Rank DETR achieves 49.6 AP (12 epochs), while Stable DINO achieves 50.4 AP (12 epochs).
- The definition of ranking in DETR is not clear enough. What criterion is the ranking based on? The IoU or the predicted confidence. This should be clearly pointed out at the very beginning.
- The relation between the proposed method and IoU-related methods is not clear enough. Section 2 should explain their differences against IoU-related methods, as well as connections, in more details.


**Questions:**

- Eqn. 2 is confusing. What does Sort(A, B) perform? Sorting B and duplicate the sorting results onto A?
- In Eqn. 2, positional embedding is irrelevant to the rank of a predicted results, but determined by the coordinates. Since the ranking (sorting) does not change the coordinates, how does the ranking impact the positional embedding? If there is no impact, why do you enforce ranking in Eqn. 2?

**Limitations:**

The authors have  discussed the limitations.

---

> ### Author Rebuttal · Authors · 2023-08-10
>
> ## To Reviewer JDo7
>
> We thank the reviewer for the careful reviews and constructive suggestions. We answer the questions as follows.
>
> > **"Two (out of four) major components ..."**
>
> A:
> 👉 First, we summarize their mathematical formulations as follows:
>
> |   method  | classification loss modification  | matching cost modification |
> | :-------------  | :------------------:  |  :-------------:  |
> | Stable-DINO | change the classification target for positive query to: $\mathrm{IoU}^2$ | modulate the classification prediction with GIoU scores: $\mathrm{p}\times(\frac{\mathrm{GIoU}+1}{2})^{0.5}$ |
> | Align-DETR  | change the classification target for positive query to:$\mathrm{p}^{0.25}\times\mathrm{IoU}^{0.75}$ | no change |
> | Rank DETR   | change the classification target for positive query to: $\frac{\mathrm{GIoU}+1}{2}$ |  replace the original one with $\mathrm{p} \cdot {\text{IoU}}^{4}$ |
>
> According to the above formulations, we can see that the key difference is at the modification to the original matching cost, which consists of classification cost, $\ell_1$ loss, and GIoU loss.
>
> 👉 Second, we agree that the improvement brought by these two components seem smaller than the similar ones in the competing methods when using the DINO-DETR as the baseline. We attempt to explain the possible reasons in the general response.
>
> ---
>
> > **"More importantly, though the ..."**
>
> A: Great point!
>
> 👉 First, we want to clarify that **the mentioned the 50.4 AP (12 epochs) of Stable-DINO relies on using another two techniques including: (i) NMS and (ii) Memory fusion**, where the results are summarized in Table 6 of Stable-DINO paper and these two techniques bring additional +0.4 AP gains. Therefore, **the actual improvement brought by the IoU-related loss (and matching) in Stable-DINO is 1.0 AP gain**.
>
> 👉 Second, we have provided more comparison results in the general response. Align-DETR and Stable-DINO seem to use an improved DINO-DETR baseline with AP=49.4 (vs. ours: AP=48.7).
>
> ---
>
> > **"The definition of ranking in DETR ..."**
>
> A: Thanks for pointing out this issue and we address your concerns as follows:
>
> 👉 **Ranking definition**: ranking in DETR represents **the ranking order of the object query (and the associated bounding box predictions)**. The ranking order of object query is impoxrtant because the final predictions of modern DETR methods (H-DETR, DINO-DETR) are generated by sorting the bounding box predictions based on the classification confidence scores in descending order by default.
>
> 👉 **Criterion of ranking**: the modern DETR methods (H-DETR, DINO-DETR) choose the default classification scores (agnostic to the localization precision) as the ranking criterion. In this work, we choose the **GIoU-aware classification scores** (predicted by the classification head) as the ranking criterion. These GIoU-aware classification scores of the positive samples are supervised by its normalized GIoU scores ($(\mathrm{GIoU}(\hat{\mathbf{b}}, \mathbf{b}) + 1)/2$) with the matched ground-truth box.
>
> ---
>
> > **"The relation between the proposed ..."**
>
> A: Thanks for your valuable suggestions! We would like to include the following discussion in the final revision following your comments. We address your concerns on the connections and differences between our method and the IoU-related methods as follows:
>
> 👉 Connections: the most significant connection exists between the IoU-related methods and our work, alongside Stable-DINO and Align-DETR. A pivotal similarity lies in our shared objective, which centers around **addressing the mis-alignment between classification scores and localization accuracy.**
>
> 👉 Differences:
>
> - **Designed for different detectors**: the previous IoU-related methods focus on improving the ranking scheme for conventional object detectors such as FPN, Cascade R-CNN, FCOS, and ATSS. Our work focuses on improving the ranking design for modern DETR-based object detecters considering H-DETR/DINO-DETR already achieves stronger results.
>
> - **DETR-oriented innovations**: our innovation lies in the domain of DETR-specific designs, specifically our pioneering rank-oriented architecture for DETR methods. These elements are ingeniously crafted to cater to the intricate task of investigating the ranking order of object queries within DETR-based object detectors.
>
> ---
>
> > **"Eqn. 2 is confusing..."**
>
> A: Thanks for pointing out this issue! Sort(A, B) means sorting the elements within A according to the decreasing order of the elements within B. In the Eqn. 2, we use $\operatorname{Sort}({\mathcal{B}}^{l}, \mathcal{P}^{l})$ to represent sorting the order of bounding box predictions $\mathcal{B}^{l}$ according to the decreasing order of their corresponding classification scores $\mathcal{P}^{l}$.
>
> We implement the Sort() function with the combination of two functions including torch.argsort() and torch.gather(). First, we use torch.argsort() to get the indices of the sorted probability predictions $\mathcal{P}^{l}$ and torch.argsort() is not differentiable. Second, we use torch.gather() to gather the values of $\mathcal{B}^{l}$ according to the output of torch.argsort().
>
>
> ---
>
> > **"In Eqn. 2, positional embedding ..."**
>
> A:  (i) Yes, positional embedding is determined by the coordinates (bounding box predictions). (ii) The ranking of the positional embedding essentially places the bounding box predictions associated with higher GIoU-aware classification scores in the top-ranked positions. (iii) The reason for enforcing ranking in Eqn. 2 is to **keep the order of the positional embedding aligned with the order of the ranked content query** (shown in Eqn. 3) as we need to send the combination of rank-aware positional query embedding and content query embedding into the subsequent Transformer decoder layers.

---

> > ### Author Response · Authors · 2023-08-17
> > **Looking forward to hearing the feedback from JDo7**
> >
> > We sincerely value your dedicated guidance in helping us enhance our work. We are eager to ascertain whether our responses adequately address your primary concerns, particularly in relation to the comparisons with Stable DINO and Aligned DETR. We would be grateful for the opportunity to provide any needed further feedback.

---

> > ### Comment · Reviewer_JDo7 · 2023-08-17
> > **Thanks for the authors' rebuttal**
> >
> > The authors have clarified most unclear statements and explained their differences and connections with recent IoU-based DETR methods. Though the proposed method shares similarities with some recent IoU-based DETR methods (e.g., Stable DINO and Align DETR), the reviewer is convinced that these works are actually concurrent, and thus has no more doubt on their contributions. Moreover, during rebuttal, the authors have showed that compared with Stable DINO, their method can actually achieve comparable results on the popular baseline DINO and even slightly higher accuracy on a more recent method (H-DETR).Therefore, I would like to change my rating to "weak accept".

---

### Official Review · Reviewer_m522 · 2023-07-07

**Soundness:** 3 good
**Presentation:** 3 good
**Contribution:** 2 fair
**Rating:** 5
**Confidence:** 4

**Summary:**

This paper proposes Rank-DETR for image object detection. The key contributions of Rank-DETR include (1) a rank-oriented architecture design, which comprises a rank-adaptive classification head and query rank layer to ensure lower FP and FN in predictions; (2) a rank-oriented loss and matching design, which introduces GIoU-aware classification loss and high-order matching cost to boost the AP under high IoU thresholds. Experiments on COCO demonstrate each component can contribute to the overall performance and align the design expedition, and RANK-DETR outperforms the current SOTA detector DINO.

**Strengths:**

1. The paper is well written, neatly written, clean, and easy to follow.
2. The motivation of the paper is clear, 1. Rank-oriented Architecture Design: ensure lower FP and FN 2. Rank-oriented Matching Cost and Loss: boost the AP under high IoU thresholds. Experiments demonstrate the proposed component can align well with the design intent.
3. The experiments are very solid, it compares two SOTA query-based detectors H-DETR and DINO, and outperform them. The ablation study thoroughly analyzes each component's effect and performance contribution to demonstrate its effectiveness.

**Weaknesses:**

1. Some module designs are not very novel. (1) Rank-adaptive Classification Head learns a class-aware and input-independent logits vector to model the class distribution and calibrate each query's score predictions. Such technique is well used in other fields like long-tailed classification/detection, and few-shot classification/detection. (2) GIoU-aware Classification Loss is very similar to Veri-Focal-Loss, except the modulating factor of positive classification loss is $t - \hat{p}[l]$ instead of $t$, and such a minor modification leads to a slight performance boost, $~0.3mAP$.

2. The Query Rank Layer design decouples content and positional query. The content query is constructed from MLP mapping of the fusing of a static query $C_l$ and output of the last decoder layer $O_l$. The positional query is constructed from PE encoding of ranked predicted boxes. This is a new design of query initialization, it will be interesting to compare it with other query initialization methods in DETR, Deformable-DETR, and DINO, but I can't find such comparisons.

3. There is no analysis or comparison of parameter FLOPs and computation cost (training & testing speed).

**Questions:**

Please refer to the weakness section.

---

> ### Author Rebuttal · Authors · 2023-08-10
>
> ## To Reviewer m522
>
> We thank the reviewer for the careful reviews and constructive suggestions. We answer the questions as follows.
>
> ---
>
> > **"Some module designs are not very novel. (1) Rank-adaptive Classification Head learns a class-aware and input-independent logits vector to model the class distribution and calibrate each query's score predictions. Such technique is well used in other fields like long-tailed classification/detection, and few-shot classification/detection. (2) GIoU-aware Classification Loss is very similar to Veri-Focal-Loss, except the modulating factor of positive classification loss is $t-\hat{\mathbf{p}}[l]$ instead of $t$, and such a minor modification leads to a slight performance boost, $0.3$ mAP."**
>
> A: Great point!  We acknowledge the similarity of the mentioned module designs to previous techniques. However, our simple designs already showcase significant potential in exploring rank-oriented concepts for DETR-based object detectors. We hope that our efforts can inspire more advanced rank-oriented designs. Last, we welcome any further valuable suggestions to continue improving these module designs. Your insights are greatly appreciated!
>
> ---
>
> > **"The Query Rank Layer design decouples content and positional query. The content query is constructed from MLP mapping of the fusing of a static query $C_l$ and output of the last decoder layer $O_l$. The positional query is constructed from PE encoding of ranked predicted boxes. This is a new design of query initialization, it will be interesting to compare it with other query initialization methods in DETR, Deformable-DETR, and DINO, but I can't find such comparisons."**
>
> A: Great point! We follow your suggestion to compre the Query Rank Layer method with the query initialization methods of DETR, Deformable-DETR, and DINO as follows:
>
> | Method          | Content Query |  Positional Query |
> |-----------------|----------|-------|
> | DETR            | iterative refine | shared across layers, learnable queries |
> | Deformable-DETR | iterative refine | shared across layers, introduced with bounding box |
> | DINO            | iterative refine | regenerated at each layer from bounding box |
> | Ours            | iterative refine + sorted embedding | regenerated at each layer from bounding box |
>
> We also ablate the proposed QRL (query initialization methods) in Deformable-DETR and DINO in 1x schedule. DETR results is not provided because it need long epoch to converge.
>
> | Method          | Backbone |  QRL  | mAP |
> |-----------------|----------|-------|-----|
> | Deformable-DETR | R50      |  ❎  | 43.4 |
> | Deformable-DETR | R50      |  ✅  | 45.0 |
> | DINO            | R50      |  ❎  | 48.7 |
> | DINO            | R50      |  ✅  | 49.3 |
>
> According to the above comparison results, we can see the Query Rank Layer consistently outperforms other query initialization methods.
>
> ---
>
> > **"There is no analysis or comparison of parameter FLOPs and computation cost (training & testing speed)."**
>
> A: Great point! We follow your suggestion to provide the comparison of parameter FLOPs and computation cost (training & testing speed) as follows:
>
> | Method      | Backbone | Params(M) | FLOPs(G) | Training Cost (min) | Testing FPS (img/s) | mAP  |
> |-------------|----------|-----------|----------|---------------------|---------------------|------|
> | H-DETR      | R50      | 47.56     | 280.30    |   69.8 min  | 19.2 | 48.7 |
> | Rank-H-DETR | R50      | 49.10     | 280.60    |   71.8 min  | 19.0 | 50.0 |
>
>
> We conducted testing and training cost evaluations utilizing the RTX 3090 GPU. The outcomes reveal a noteworthy enhancement in detection performance through our proposed method, with a marginal increase in FLOPs and inference latency.

---

> > ### Comment · Reviewer_m522 · 2023-08-18
> >
> > Thanks for the author's feedback, which resolved all of my concerns, I would keep my initial rating "bordering accept".

---

### Author Rebuttal · Authors · 2023-08-10

## To AC and All Reviewers

We thank all the reviewers for their careful reviews and constructive suggestions. These constructive feedbacks has significantly contributed to the improvement of our paper. We are glad to find the initial ratings of three reviewers (Reviewer m522, Reviewer kDha, and Reviewer vAjC) are positive.

Above all, we attempt to address the major concern on the differences and comparison with Stable-DINO and Align-DETR, from the following aspects:

> **Stable-DINO and Align-DETR are con-current works**

We would like to highlight that Stable-DINO was made available on arXiv on **10th April 2023**, while Align-DETR became accessible on **15th April 2023**. It is important to note that both of these works were not formally published at the time of this submission on **11th May 2023**. We have duly mentioned this aspect in the related work section of our paper, where we discuss the relevance and relationship of our approach to these contemporaneous works.

> **Different motivation and insight**

The motivation of Stable-DINO is to **address the unstable matching problem across different decoder layers** and the motivation of Align-DETR is to **address the misalignment between classification score and localization precision**. We acknowledge the motivation of our rank-oriented loss and matching design is close to the Align-DETR.

Different from both of them, the motivation of our rank-oriented architecture design is to **prompt positive predictions and suppress the negative ones to ensure lower false positive rates**.

> **Stable-DINO and Align-DETR introduced additional technical improvements**

(1) Stable-DINO further improves DINO-DETR with other designs. According to Table 6 in Stable-DINO paper, **first, applying NMS during evaluation brings +0.2 gains (49.0->49.2). Second, The combination of dense memory fusion and NMS brings +0.4 gains (49.0->49.4). Third, The combination of position-supervised loss and position-modulated matching cost brings the other +1.0 gains (49.4->50.4)**. According to Figure 7 in Stable-DINO paper, we notice that the memory fusion method, which concatenates the 24x (4 scales x 6 encoder layers) multi-scale encoder feature maps with the 4x multi-scale backbone feature maps followed by linear projection and normalization, will bring additional computation overhead during both training and evaluation, e.g., during evaluation, the GFLOPs increases from 289.90 G to 300.12 G.

(2) Align-DETR further improves DINO-DETR with **a mixed matching strategy** (introduce more positive samples) and **a prime sample weighting scheme** (down-weight the loss for low-quality positive samples). According to Table 5 and Table 8 in Align-DETR paper, these two techniques bring 0.2 gains (50->50.2).

Different from both of them, we have proposed to improve the DETR-based methods from a novel aspect, i.e., rank-oriented architecture design, which brings +0.9 gains over the baseline. We further clarify a possible misunderstanding of Reviewer kDha (**the main boost originates from the GIoU-aware loss**). We summarize some key results (from Table 4 in the main paper) in the following Table for reference. Accordingly, we can see that the proposed rank-oriented architecture design boosts the baseline from 48.2 to 49.1 while rank-oriented matching and loss design boosts the baseline from 48.2 to 49.5, respectively.

|  Rank-oriented architecture design (RCH + QRL)   | Rank-oriented loss and matching design (GCL + HMC) | mAP  | AP50  | AP75  |
| --- | --- | ---- |---- | ---- |
| ❎ | ❎ | 48.2 | 66.4 | 52.6 |
| ✅ | ❎ | 49.1 | 67.2 | 53.5 |
| ❎ | ✅ | 49.5 | 67.3 | 54.0 |
| ✅ | ✅ | 50.0 | 67.5 | 54.7 |

> **Concern about the smaller improvements than Stable-DINO and Align-DETR**

- First, we would like to highlight that Stable-DINO and Align-DETR **have tuned the hyperparameters and conducted all ablation experiments based on DINO-DETR**. For the experiments based on DINO-DETR, our goal is to **verify the generalization ability** and we simply **use the same hyperparameters as the experiments based on H-DETR without any tuning**. An interesting observation is that **our Rank-DETR (AP=50.0) significantly outperforms Stable-H-DETR (AP=49.2) when using the H-DETR as baseline**, where we notice that the H-DETR baseline of Stable-H-DETR is even stronger than our baseline: 48.6 vs. 48.2.

|   method  | mAP  | AP50  | AP75  |
| -------------  | ---- |---- | ---- |
| H-DETR (Stable-DINO reproduce) | 48.6 | - | - |
| Stable-H-DETR | 49.2 | - | - |
| H-DETR (our reproduce) | 48.2 |  66.4 | 52.9 |
| Rank H-DETR   | 50.0 |  67.5 | 54.7 |

- Second, to address the concerns on results over DINO-DETR, we report the detailed comparison results to Stable-DINO by using the additional special tricks as follows:

|   method   | NMS & Memory fusion | mAP  | AP50  | AP75  |
| ------------- | --- | ---- |---- | ---- |
| DINO-DETR (our reproduced baseline)  | ❎  | 48.7 | 66.1 | 52.9 |
| Rank DINO-DETR    |  ❎  | 49.6 | 67.0 | 54.7 |
| Rank DINO-DETR    |  ✅  | 50.4 | 67.9 | 55.3 |

Besides, we also attempt to leverage these techniques to improve our Rank H-DETR and we further achieve even stronger results than Stable-DINO, i.e., AP=$50.8$.

- Last, we also would like to point out another possible reason, Align-DETR (https://github.com/FelixCaae/AlignDETR) chooses the DINO baseline reproduced based on the detrex codebase (https://github.com/IDEA-Research/detrex/tree/main/projects/dino). According to their README, we find that the reproduced DINO-DETR baseline already achieves 49.4 in detrex implementation. We would like to reimplement our method on the detrex codebase if it is necessary.

👉 In conclusion, we earnestly hope that our work won't be rejected solely based on lacking direct competitive comparisons with concurrent endeavors. We firmly believe that our contribution can **offer new insights to the community on how to better exploit the ranking information for DETR-based object detectors**.

---

> ### Author Response · Authors · 2023-08-12
> **Looking forward to hearing the response from all Reviewers**
>
> We sincerely value the reviewers for your thorough assessments and invaluable suggestions that have significantly enhanced our submission.
>
> We eagerly anticipate additional suggestions and improve our work from your further invaluable comments.

---

### Decision · Program_Chairs · 2023-09-21

**Decision:**

Accept (poster)

**Comment:**

The paper proposes Rank-DETR, a DETR-based object detector incorporating rank-oriented designs at both architecture and loss levels. The rank-adaptive classification head and query rank layer introduce rank awareness into the architecture. The GIoU-aware classification loss and high-order matching cost integrate ranking into the loss and matching. Experiments demonstrate consistent improvements over baselines, aligning with design motivations. The reviewers rate the paper positively overall. ACs concur and recommend acceptance, considering the technically solid contribution despite limited direct comparison to very recent concurrent works addressing similar ideas.